# Irrigation Management Strategies to Enhance Forage Yield, Feed Value, and Water-Use Efficiency of Sorghum Cultivars

**DOI:** 10.3390/plants12112154

**Published:** 2023-05-30

**Authors:** Amir Ghalkhani, Farid Golzardi, Azim Khazaei, Ali Mahrokh, Árpád Illés, Csaba Bojtor, Seyed Mohammad Nasir Mousavi, Adrienn Széles

**Affiliations:** 1Department of Agronomy and Plant Breeding, Karaj Branch, Islamic Azad University, Karaj P.O. Box 31485-313, Iran; amirghalkhani@yahoo.com; 2Seed and Plant Improvement Institute, Agricultural Research, Education and Extension Organization (AREEO), Karaj P.O. Box 31585-4119, Iran; f.golzardi@areeo.ac.ir (F.G.); a.khazaei@areeo.ac.ir (A.K.); a.mahrokh@areeo.ac.ir (A.M.); 3Institute of Land Use, Engineering and Precision Farming Technology, University of Debrecen, 138 Böszörményi St., 4032 Debrecen, Hungary; illes.arpad@agr.unideb.hu (Á.I.); bojtor.csaba@agr.unideb.hu (C.B.); szelesa@agr.unideb.hu (A.S.); 4Department of Plant, Food and Environmental Sciences, Faculty of Agriculture, Dalhousie University, Truro, NS B2N 5E3, Canada

**Keywords:** deficit irrigation, drip irrigation, dry matter yield, forage quality, furrow irrigation, metabolizable energy, water saving

## Abstract

Water scarcity is a major obstacle to forage crop production in arid and semi-arid regions. In order to improve food security in these areas, it is imperative to employ suitable irrigation management techniques and identify drought-tolerant cultivars. A 2-year field experiment (2019–2020) was conducted in a semi-arid region of Iran to assess the impact of different irrigation methods and water deficit stress on forage sorghum cultivars’ yield, quality, and irrigation water-use efficiency (IWUE). The experiment involved two irrigation methods, i.e., drip (DRIP) and furrow (FURW), and three irrigation regimes supplied 100% (I_100_), 75% (I_75_), and 50% (I_50_) of the soil moisture deficit. In addition, two forage sorghum cultivars (hybrid Speedfeed and open-pollinated cultivar Pegah) were evaluated. This study revealed that the highest dry matter yield (27.24 Mg ha^−1^) was obtained under I_100_ × DRIP, whereas the maximum relative feed value (98.63%) was achieved under I_50_ × FURW. Using DRIP resulted in higher forage yield and IWUE compared to FURW, and the superiority of DRIP over FURW increased with the severity of the water deficit. The principal component analysis indicated that, as drought stress severity increased across all irrigation methods and cultivars, forage yield decreased, while quality increased. Plant height and leaf-to-stem ratio were found to be suitable indicators for comparing forage yield and quality, respectively, and they showed a negative correlation between the quality and quantity of forage. DRIP improved forage quality under I_100_ and I_75_, while FURW exhibited a better feed value under the I_50_ regime. Altogether, in order to achieve the best possible forage yield and quality while minimizing water usage, it is recommended to grow the Pegah cultivar and compensate for 75% of soil moisture deficiency using drip irrigation.

## 1. Introduction

Irrigation is a crucial factor in crop productivity in many regions of the world [1]. The combination of water scarcity, climate change, severe droughts, and increasing demand for water from various sectors has significantly diminished crop yields, threatened food security, compromised the sustainability of agriculture, and limited access to irrigation in numerous regions [2,3]. Various strategies have been developed to mitigate the adverse effects of limited irrigation on crops, including the cultivation of drought-tolerant species and cultivars [4,5], optimized irrigation scheduling [6,7], and the use of suitable and water-saving irrigation methods [2,8,9]. For instance, growing drought-tolerant crops such as sorghum can improve water-use efficiency and reduce the impact of water stress on yield [1,10]. The high yield and drought tolerance of sorghum make it a crucial crop for providing livestock feed and human food in arid and semi-arid regions [11]. For this reason, understanding the impact of irrigation strategies on the physiological behavior of this crop is critical for implementing more efficient water management in these regions.

Research studies indicate that the choice of cultivar significantly impacts sorghum’s forage yield, quality, and water-use efficiency [12,13]. Variations in drought tolerance, growth rate, leaf-to-stem ratio, and leaf area index of different cultivars may affect the crop’s water-use efficiency [14,15]. Sorghum genotypes with higher drought tolerance and growth rates have the potential to produce more forage per unit of water, leading to improved water-use efficiency and sustainable production [13]. These genotypes may also exhibit greater leaf area index and leaf-to-stem ratio, improving light interception and photosynthesis rates, ultimately resulting in higher yields and better-quality forage [16,17].

Iran is experiencing a severe water scarcity crisis, with a per capita water availability value significantly below the global average [18]. Factors such as high water consumption, unsustainable agricultural practices, and population growth have worsened the situation [19]. This crisis has led to severe water shortages in various regions of Iran, impacting the livelihoods of millions of people and posing risks to food security and socio-economic development [20]. In regions with limited rainfall or water resources, irrigation is often necessary to maintain yields and quality of forage sorghum [1,21]. 

Effective irrigation management can significantly impact the forage production potential and water-use efficiency of sorghum. [2]. Studies have shown that different irrigation methods, including surface, sprinkler, and drip irrigation, have different effects on crop yield and yield quality [9,22,23]. The applied irrigation method significantly affects water distribution in the soil profile and water losses, with crucial implications for sorghum yield and quality of forage [2,24]. While furrow irrigation is a traditional method for crop irrigation in various regions of the world, it can lead to water losses due to runoff and evaporation despite providing water to crops uniformly [2,25]. However, drip irrigation is a more precise method of delivering water directly to the crop roots, thereby reducing water loss and increasing water-use efficiency [5]. Studies have shown that drip irrigation can increase the water-use efficiency of crops by up to 30–60% compared to flood or furrow irrigation [23,25,26]. 

Deficit irrigation is a widely implemented strategy in various parts of the world to conserve water and increase water-use efficiency in agriculture [3]. Improved water-use efficiency was reported as a result of deficit irrigation techniques for various plants [27,28,29,30]. Nevertheless, this practice can have a significant impact on the yield and quality of forage crops, particularly sorghum, which are crucial components of animal feed and food security [1,21]. The effects of limited irrigation on forage sorghum can vary based on several factors, such as the amount and frequency of irrigation, the crop’s developmental stage, and the plant’s genetics [31,32]. Under limited irrigation, plants may undergo water stress, resulting in reduced water uptake, wilting, reduced photosynthesis and growth, and ultimately a decline in the yield of forage crops [3,33]. Studies have shown that drought stress can modify gene expression associated with water balance and stress response, ultimately leading to the activation of defense mechanisms [34,35]. Furthermore, limited irrigation can affect the water-use efficiency of forage sorghum, which is a critical factor in sustainable agriculture [1].

Sorghum plays a critical role in providing animal feed and maintaining food security. Alongside the growing demand for forage crops and the limited availability of water in various regions around the world, it is essential to understand the effects of irrigation management techniques on the forage yield and quality of sorghum cultivars. It is of vital importance to gain a comprehensive understanding of the influence of water deficit stress and irrigation methods on forage production potential and water-use efficiency in devising efficient strategies to mitigate the adverse effects of drought stress and achieve sustainable forage production. Therefore, this study aims to evaluate the performance of forage sorghum cultivars under different irrigation regimes and methods to determine optimal management practices for maximizing sorghum’s forage production, feed value, and water-use efficiency. The findings of this study will contribute significantly to the existing body of knowledge and inform the development of efficient strategies to mitigate the adverse effects of drought stress and achieve sustainable forage production in regions facing water scarcity.

## 2. Results

### 2.1. Green Herbage and Dry Matter Yield

Green herbage yield (GHY) and dry matter yield (DMY) were significantly affected by the irrigation method, irrigation regime, and their interaction. Additionally, the interaction effects of irrigation regime and irrigation method with cultivar also had a significant impact on GHY. In the second year of the experiment, yields were higher than the first year, with GHY and DMY being 4% and 5% higher, respectively. Drip irrigation increased DMY and GHY by 11–12% compared to FURW. Moderate and severe drought stress decreased GHY by 17% and 33% and DMY by 15% and 31%, respectively, compared to full irrigation conditions. The highest GHY and DMY (125.7 and 27.24 Mg ha^−1^, respectively) were obtained under I_100_ × DRIP. Under DRIP, moderate and severe drought stress reduced GHY by 14% and 29%, respectively, and DMY by 12% and 27%, respectively, while under FURW, the reduction was 19% and 38% for GHY and 17% and 35% for DMY. The cultivar Pegah exhibited maximum GHY under DRIP and minimum GHY under FURW. The examined genotypes reacted differently to the irrigation method, with the hybrid Speedfeed under FURW and the cultivar Pegah under DRIP having more forage yield. The highest and lowest GHY (125.49 and 80.17 Mg ha^−1^, respectively) were obtained by the cultivar Pegah under I_100_ and I_50_, respectively. The cultivar Pegah was more sensitive to deficit irrigation than the hybrid Speedfeed, with moderate and severe drought stress decreasing GHY by 19% and 36% in the cultivar Pegah and 14% and 30% in the hybrid Speedfeed, respectively (Table 1, Table 2, Table 3, Table 4, Table 5 and Table 6).

### 2.2. Irrigation Water-Use Efficiency

The irrigation method, irrigation regime, and their interaction significantly affected irrigation water-use efficiency (IWUE), as did the interaction effects of irrigation regime and irrigation method with cultivar (Table 1, Table 3, Table 5 and Table 6). DRIP resulted in a 64% improvement in IWUE compared to FURW, and limited irrigation led to an increase in IWUE, with moderate and severe drought stress leading to a 14% and 40% improvement, respectively (Table 1). The highest IWUE value was achieved under I_50_ × DRIP (Table 3). DRIP was more efficient with increasing water limitation intensity, with IWUE under I_100_, I_75_, and I_50_ by DRIP being 54%, 63%, and 73% higher than FURW, respectively (Table 3). Both irrigation methods showed a significant increase in IWUE as drought stress intensified, with DRIP exhibiting a greater change rate. Under DRIP, moderate and severe drought stress increased IWUE by 17% and 46%, respectively, while under FURW it increased by 10% and 30%, respectively (Table 3). The hybrid Speedfeed demonstrated superior IWUE under FURW compared to the cultivar Pegah, but there was no significant difference between the two cultivars under DRIP (Table 5). As drought stress intensity increased, the IWUE of both cultivars improved, although the change rates were not identical. The IWUE of the hybrid Speedfeed under I_75_ and I_50_ increased by 18% and 46%, respectively, compared to I_100_, while the increase in the cultivar Pegah was 11% and 34%, respectively (Table 6).

### 2.3. Plant Height and Leaf-To-Stem Ratio

Plant height (PLH) was significantly influenced by the irrigation regime, irrigation method × irrigation regime, irrigation method × cultivar, and irrigation regime × cultivar (Table 1, Table 3, Table 5 and Table 6). The leaf-to-stem ratio (L:S) was also significantly affected by irrigation regime, cultivar, and irrigation method × irrigation regime (Table 1 and Table 3). Deficit irrigation led to a decrease in PLH and an increase in L:S. Specifically, under I_75_ and I_50_, PLH decreased by 8% and 22%, respectively, while L:S increased by 3% and 7%, respectively, compared to I_100_ (Table 1). The Pegah cultivar had a higher L:S than the hybrid Speedfeed (Table 1). The maximum PLH (219 cm) was recorded under I_100_ × DRIP, whereas the lowest PLH (166.5 cm) and the highest L:S (0.637) were obtained under I_50_ × FURW (Table 3). The PLH of the cultivar Pegah was lower than that of the hybrid Speedfeed under both irrigation methods, although the difference between the two cultivars was relatively smaller under FURW (Table 5). The highest PLH (221.5 cm) was obtained in the hybrid Speedfeed under I_100_, while the lowest PLH (161.5 cm) was recorded in the cultivar Pegah under I_50_ (Table 6). The response of the studied cultivars to the irrigation regime varied, with moderate and severe drought stress leading to a 10% and 21% reduction in PLH, respectively, for the hybrid Speedfeed, while the reductions for the cultivar Pegah were 7% and 23%, respectively (Table 6).

### 2.4. Protein Yield and Content

The crude protein yield (CPY) was significantly impacted by irrigation method, cultivar, irrigation method × irrigation regime, irrigation method × cultivar, and irrigation regime × cultivar, as well as irrigation regime (Table 1, Table 3, Table 5 and Table 6). Crude protein content (CPC) was also significantly influenced by irrigation regime, irrigation method × irrigation regime, irrigation method × cultivar, and irrigation method × irrigation regime × cultivar, as well as cultivar (Table 2, Table 4 and Table 5). The CPY and CPC were found to be significantly higher under DRIP compared to FURW, with an increase of 16% and 3%, respectively (Table 1 and Table 2). Drought stress affected the CPY and CPC differently, with CPY decreasing by 12% and 27% under I_75_ and I_50_, respectively, while CPC increased by 4% and 6%. The cultivar Pegah showed higher CPC and CPY values than the hybrid Speedfeed, with an increase of 21% and 17%, respectively (Table 1 and Table 2). The highest CPY (2362 kg ha^−1^) was obtained under I_100_ × DRIP, whereas the lowest CPY (1480 kg ha^−1^) was recorded under I_50_ × FURW (Table 3). The highest and lowest CPC (91.5 and 83.2 g kg^−1^, respectively) were found under I_50_ × DRIP and I_100_ × FURW, respectively (Table 4). Drought stress caused a decrease in CPY under both irrigation methods, but the decrease was greater under FURW. DRIP was found to be more effective with increasing drought stress intensity (Table 3). The maximum CPY (2345 kg ha^−1^) was obtained in the cultivar Pegah under DRIP, whereas the CPY of hybrid Speedfeed showed no significant response to the irrigation method (Table 5). The CPY of the cultivar Pegah was more sensitive to deficit irrigation, with reductions of 15% and 30% under I_75_ and I_50_, respectively, compared to 8% and 23% for the hybrid Speedfeed (Table 6). The maximum and minimum CPY (2489 and 1551 kg ha^−1^) were recorded for the cultivar Pegah under I_100_ and the hybrid Speedfeed under I_50_, respectively (Table 6). The highest CPC (101.1 g kg^−1^) was obtained in the cultivar Pegah under I_50_ × DRIP, whereas the lowest CPC (75.8 g kg^−1^) was recorded in the hybrid Speedfeed under I_100_ × FURW (Figure 1). Drip irrigation increased CPC in both cultivars, but its effect was more pronounced for the cultivar Pegah. Moreover, drought stress increased CPC across all irrigation methods and cultivars (Figure 1).

### 2.5. Forage Digestibility

Dry matter digestibility (DMD) and organic matter digestibility (OMD) were significantly influenced by the irrigation regime, cultivar, and irrigation method × irrigation regime, with the interaction effect of irrigation method × irrigation regime × cultivar on OMD being significant (Table 2 and Table 4). Digestible dry matter yield (DDMY) was also significantly affected by irrigation method, irrigation regime, irrigation method × irrigation regime, irrigation method × cultivar, and irrigation regime × cultivar (Table 1, Table 3, Table 5 and Table 6). Drip irrigation led to a 12% increase in DDMY compared to FURW (Table 3 and Table 4). Limited irrigation resulted in decreased DDMY and increased DMD and OMD. Moderate and severe drought stress caused a 14% and 30% decrease in DDMY, accompanied by an increase in DMD and OMD. The DMD and OMD of the cultivar Pegah were approximately 4% higher than those of the hybrid Speedfeed, but there was no significant difference in DDMY between the two cultivars (Table 1 and Table 2). The highest DDMY (16,005 kg ha^−1^) was obtained under I_100_ × DRIP, whereas the lowest DDMY (9984 kg ha^−1^) and the highest DMD and OMD (597.4 and 562.2 g kg^−1^, respectively) were recorded under I_50_ × FURW (Table 3 and Table 4).

Moderate and severe stress reduced DDMY by 12% and 26% in DRIP and 17% and 33% in FURW, respectively. Drip irrigation was more effective with increasing drought stress intensity (Table 3). Drought stress had a more severe impact on DMD and OMD under FURW compared to DRIP (Table 4). The DDMY of different cultivars responded differently to the irrigation method. The hybrid Speedfeed under FURW and the cultivar Pegah under DRIP showed higher DDMY. The maximum and minimum DDMY (14,356 and 12,127 kg ha^−1^) were obtained in the cultivar Pegah under DRIP and FURW, respectively (Table 5). The cultivar Pegah was more sensitive to limited irrigation, with its DDMY decreasing by 17% and 33% under I_75_ and I_50_, respectively, while the hybrid Speedfeed’s DDMY decreased by 12% and 27%, respectively (Table 1). The cultivar Pegah under I_50_ × FURW had the maximum OMD (571.4 g kg^−1^), whereas the hybrid Speedfeed under I_100_ × FURW had the minimum OMD (539.8 g kg^−1^) (Figure 2). DRIP showed higher OMD under I_100_ and I_75_, whereas FURW was superior under I_50_. Both cultivars showed an increase in OMD with the increasing drought stress intensity, with the rate of these changes being higher under FURW than DRIP. Additionally, the cultivar Pegah exhibited a stronger response to drought stress compared to the hybrid Speedfeed (Figure 2).

### 2.6. Fiber Content

Based on the study, it was concluded that the irrigation regime and cultivar had a significant effect on acid detergent fiber (ADF) and neutral detergent fiber (NDF), with NDF also being significantly affected by the year and irrigation regime × cultivar. The second year of the study had higher NDF than the first year. As drought stress severity increased, the ADF and NDF decreased significantly (Table 2). Compared to full irrigation, I_75_ and I_50_ led to reductions in ADF and NDF. The cultivar Pegah had lower ADF and NDF than the hybrid Speedfeed (Table 2). The highest ADF and NDF (394.9 and 604.1 g kg^−1^, respectively) were obtained under I_100_ × FURW, whereas the lowest ADF and NDF (374.4 and 563.8 g kg^−1^, respectively) were recorded under I_50_ × FURW (Table 4). The rate of change in ADF and NDF due to drought stress was higher under FURW than under DRIP. Under I_100_ and I_75_, DRIP had the lowest ADF and NDF, while under I_50_, FURW had the lowest values (Table 4). The mean comparison of the irrigation regime × cultivar interaction revealed that the highest NDF (614.8 g kg^−1^) was recorded for the hybrid Speedfeed under I_100_, whereas the lowest NDF (562.6 g kg^−1^) was found for the cultivar Pegah under I_50_ (Table 6).

### 2.7. Forage Intake and Feed Value

The study found significant effects of year, irrigation regime, cultivar, and irrigation method × irrigation regime on dry matter intake (DMI) and relative feed value (RFV). The irrigation regime × cultivar had a significant effect on RFV (Table 2, Table 4 and Table 6). The first year resulted in higher RFV and DMI by 1.5% and 1.4%, respectively, when compared to the second year (Table 2). Although the irrigation method did not significantly affect the RFV and DMI, both traits increased significantly with increasing drought stress intensity. Moderate and severe drought stress resulted in an increase of 2% and 5% in DMI and 3% and 7% in RFV, respectively, compared to I_100_. The cultivar Pegah had higher DMI and RFV by 4% and 8%, respectively, than the hybrid Speedfeed (Table 2). The maximum DMI and RFV (21.30 g kg^−1^ body weight and 98.63%, respectively) were obtained under I_50_ × FURW, whereas the minimum DMI and RFV (19.88 g kg^−1^ body weight and 89.6%, respectively) were observed under I_100_ × FURW (Table 4). Drought stress increased DMI and RFV significantly in both irrigation methods, with the increase rate depending on the irrigation method. Moderate and severe drought stress under FURW resulted in an increase of 3% and 7% in DMI and 3% and 10% in RFV, respectively, while under DRIP the increase in DMI was 2% and 3% and RFV was 2% and 4%, respectively (Table 4). DRIP had higher DMI and RFV under I_100_ and I_75_, while FURW showed higher values of these traits under I_50_ (Table 4). Drought stress also caused a significant increase in RFV in both cultivars, with the highest and lowest values (100.48% and 86.7%, respectively) observed in the cultivar Pegah under I_50_ and the hybrid Speedfeed under I_100_, respectively (Table 6). Compared to full irrigation, I_75_ and I_50_ increased the RFV of the cultivar Pegah by 2% and 6%, respectively, while this increase was 4% and 8% for the hybrid Speedfeed (Table 6).

### 2.8. Energy Content of Forage

The study found that irrigation regime, cultivar, and irrigation method × irrigation regime significantly influenced metabolizable energy (ME) and net energy for lactation (NEL) (Table 2 and Table 4). The severity of drought stress had a significant impact on both ME and NEL, resulting in an increase in these traits. The energy content of the cultivar Pegah was approximately 6% higher than that of the hybrid Speedfeed (Table 2). The highest ME and NEL (2.051 and 1.320 Mcal kg^−1^, respectively) were achieved under I_50_ × FURW, whereas the lowest ME and NEL (1.965 and 1.266 Mcal kg^−1^, respectively) were recorded under I_100_ × FURW (Table 4). Although drought stress increased both ME and NEL in both irrigation methods, the increase was more pronounced under FURW. Specifically, under FURW, I_75_ and I_50_ led to an ME increase of 1.0% and 4.4% and an NEL increase of 0.8% and 4.3%, respectively, compared to full irrigation. In contrast, under DRIP, the ME increase was 0.7% and 1.5%, and the NEL increase was 0.5% and 1.4%, respectively. Furthermore, under I_100_ and I_75_, DRIP had higher ME and NEL, while under I_50_, FURW had superior levels of these traits (Table 4).

### 2.9. Principal Component Analysis

The results of the principal component analysis (PCA) based on a correlation matrix of the studied traits affected by experimental treatments are shown in Figure 3. The PC_1_ and PC_2_ accounted for 66.85% and 24.50% of the total changes in traits, respectively, representing a total of 91.35% (Figure 3). The PC_1_ was positively associated with RFV, L:S, DMI, ME, OMD, NEL, DMD, and CPC, while it was negatively influenced by NDF, ADF, and PLH. The PC_2_ was positively affected by CPY, GHY, DDMY, and DMY (Figure 3). The PC_1_ and PC_2_ were labeled as the forage quality and yield components, respectively. Higher values of PC_1_ indicate better quality forage, while higher values of PC_2_ suggest higher forage yield. The PCA findings indicate that, under all irrigation methods and cultivars, forage yield decreased, and forage quality improved as the intensity of drought stress increased, but the rate of change in traits was lower under DRIP compared to FURW. The cultivar Pegah generally had higher forage yield and quality than the hybrid Speedfeed (Figure 3). Forage yield increased under all irrigation regimes with DRIP compared to FURW, and forage quality improved under I_100_ and I_75_, although the forage quality of FURW was higher under I_50_ (Figure 3). The L:S exhibited a positive and significant correlation with several indicators of forage quality, including RFV, DMI, ME, OMD, NEL, DMD, and CPC, while it had a negative and significant correlation with NDF and ADF. The PLH had a positive and significant correlation with DMY and NDF (Figure 3). The L:S can serve as a representative measure of forage quality, while PLH can be a useful indicator of forage yield. The maximum feed value was observed in the cultivar Pegah under I_50_ × FURW, whereas the highest forage yield was obtained in the same cultivar under I_100_ × DRIP (Figure 3).

## 3. Discussion

### 3.1. Forage Quantity

The study’s findings suggest that DRIP is a more effective method for increasing the forage yield of sorghum compared to FURW, in accordance with previous research that has demonstrated the superiority of DRIP over other irrigation methods due to its better water management, precise control over water application, and reduced water loss through evaporation [36,37]. Similarly, Zhang et al. [22] found that DRIP resulted in the highest maize yield, while FURW generated the lowest yield. The current study shows that sorghum’s forage yield decreased due to drought stress, in line with earlier research [1,2,21]. Drought stress limits the plant’s photosynthetic capacity by reducing nutrient uptake from the soil, resulting in decreased assimilate production and ultimately leading to reduced growth and productive biomass [3,16,33]. The study’s findings indicate that DRIP is a more effective method for mitigating the negative impact of drought stress on sorghum’s forage yield compared to FURW [30]. The rate of yield decrease under DRIP was found to be relatively less pronounced due to the direct delivery of water to the root zone, which reduces water loss through evaporation and surface runoff [30,37]. Moreover, DRIP provides more uniform water distribution, helping to maintain soil moisture levels and reduce water stress on crops [38]. In addition, the study suggests that sorghum’s forage yield in the second year was higher than in the first year, possibly due to the higher average air temperature and sorghum’s C4 photosynthetic system [8]. The OP cultivar Pegah was found to be more sensitive to drought stress than the hybrid Speedfeed, possibly due to differences in their root systems [39]. Previous research has suggested that hybrids exhibit greater drought tolerance due to their more consistent and potentially stronger root systems, which is a result of selective breeding [12,40]. Such breeding imparts traits that improve the ability to cope with drought, such as the development of deeper or more expansive root structures [14,15,35].

### 3.2. Irrigation Water-Use Efficiency

This study provides concrete evidence that using DRIP instead of FURW significantly increases IWUE for sorghum, which is consistent with previous research [22,23]. In areas with water scarcity, FURW can lead to ineffective use of water resources and reduced crop yields due to water losses from evaporation and runoff [2]. In contrast, DRIP delivers water directly to the plant roots, resulting in substantial water savings and comparable crop yields to traditional irrigation methods [22,23]. Thus, DRIP is a more efficient method for areas facing water scarcity, requiring significantly less water than FURW [25], leading to higher IWUE due to minimized water loss from evaporation and runoff [2]. Furthermore, we observed in this study that increasing the intensity of drought stress improved IWUE, which could be due to the decreased transpiration rate and water loss caused by stomatal closure and reduction in leaf surface stomata [1,34]. Recent studies also confirmed the positive effects of limited irrigation and drought stress on IWUE improvement, which is consistent with our findings [1,3,21]. Our study found that both cultivars did not exhibit a significant difference in IWUE under full irrigation conditions. However, under drought stress, the hybrid Speedfeed demonstrated superior IWUE compared to the OP cultivar Pegah. The superiority of hybrids over OP cultivars under drought stress can be attributed to their ability to inherit desirable traits such as more vigorous root systems, drought tolerance, improved yield, and higher nutrient uptake efficiency from both parent plants [11,12].

### 3.3. Morphological Characteristics

Our investigation revealed that implementing deficit irrigation techniques resulted in a reduction in sorghum PLH while simultaneously increasing the L:S. This finding is consistent with Kaplan et al. [21], where sorghum–sudangrass plants achieved their tallest and shortest heights under full irrigation and severe drought stress conditions, respectively. Moreover, our study observed a positive correlation between the severity of drought stress and the L:S. Under water-stressed conditions, plants tend to undergo reduced cell expansion, which ultimately results in decreased PLH [40]. This decrease in PLH could be advantageous, particularly in regions with scarce water resources, as shorter plants require less water and could potentially use available resources more efficiently [41]. The increase in the L:S during drought stress may be attributed to limited water availability, which restricts stem growth more than leaf growth [1,16]. Allocating more resources toward leaves instead of stems during water-stressed conditions can result in a higher biomass yield [1,42]. Our investigation found that the L:S of the Pegah cultivar surpassed that of the Speedfeed hybrid, consistent with the results reported by Jahanzad et al. [31]. We also observed higher PLH of sorghum cultivars under DRIP than FURW in our study, which can be attributed to various factors such as efficient water delivery, improved nutrient uptake, reduced water stress, and better management of soil moisture in the drip irrigation method [24,25].

### 3.4. Forage Quality

The present study indicates that, in comparison to FURW, DRIP has a positive impact on the protein content of sorghum forage across all irrigation methods and cultivars examined. Drip irrigation offers numerous benefits for improving forage nutrient content [9]. By providing water and nutrients directly to the root zone, DRIP promotes nutrient uptake, plant growth, and improved nutritional value [43,44]. Moreover, DRIP reduces soil compaction, enhances root respiration, and increases soil enzyme activities and fertility, resulting in increased availability and uptake of nutrients and leading to higher feed value of forage [41,45]. Additionally, DRIP minimizes weed growth and competition, enabling forage plants to receive more nutrients and sunlight and leading to higher-quality forage [46,47]. These findings demonstrate that DRIP has significant potential as a superior method for improving forage quality [9].

Our study found that drought stress increases the nutritional value of sorghum forage by reducing its fiber content and increasing various nutritional attributes such as CPC, DMD, OMD, DMI, RFV, NEL, and ME. These changes are accompanied by a decrease in ADF and NDF, which can be attributed to changes in the leaf-to-stem ratio of the forage induced by drought stress conditions [1]. Similar results were reported by Kaplan et al. [21], who found an increase in the leaf-to-stem ratio and feed value of sorghum–sudangrass forage under water deficit stress. Previous studies have consistently shown that increasing the L:S of forage sorghum and sorghum–sudangrass leads to a reduction in fiber content and an increase in CPC, DMD, RFV, and energy content of the forage [1,48,49]. These improvements in forage quality under drought stress have been attributed to the superior nutritional value of leaves compared to stems, as demonstrated by earlier research [10,13,50]. Bhattarai et al. [32] found that drought stress increases the CPC and DMD but decreases the ADF and NDF of sorghum forage. Interestingly, an increase in PLH, resulting from higher water consumption, was associated with an increase in ADF and NDF, indicating a positive correlation between fiber content and plant height growth [13,32]. This finding is consistent with our study, which found that drought stress reduces plant height and improves forage quality [1,13].

The observed positive correlation between increased fiber content and plant height growth may be attributed to the plant’s efforts to reduce lodging, which ultimately decreases forage digestibility [13,51]. Sorghum exhibits an adaptive response to water scarcity, prioritizing leaf growth over stem growth as leaves require fewer resources and are less affected by water stress [10,21,52]. Under drought stress, sorghum increases its leaf-to-stem ratio, which indicates the plant’s allocation of resources toward leaf growth for the maintenance of photosynthesis [1,10]. This prioritization of photosynthesis over structural support is essential for the plant’s survival under water-limited conditions, emphasizing the intricate interplay between physiological and morphological adaptations of sorghum to cope with environmental challenges [2,9]. Farhadi et al. [1] and Pourali et al. [10] support this finding by observing an improvement in forage quality under water-limited conditions. The superior quality of forage in the cultivar Pegah compared to the hybrid Speedfeed can also be attributed to its relatively higher leaf-to-stem ratio. Consistent with the findings of the present investigation, Jahanzad et al. [31] also reported that the cultivar Pegah exhibited reduced ADF and NDF and enhanced DMD in comparison to the hybrid Speedfeed. Additionally, Mirahki et al. [11] found that forage derived from the cultivar Pegah had elevated RFV, NEL, and ME compared to that of the hybrid Speedfeed.

## 4. Materials and Methods

### 4.1. The Experimental Site, Design, and Treatments

The factorial split-plot experiment was conducted on the basis of a randomized complete block design with three replications at the Seed and Plant Improvement Institute, Karaj, Iran, during the 2019 and 2020 growing seasons. The experimental site, located in a semi-arid climate, experiences hot and dry summers and cold winters. Soil properties of the site are presented in Table 7, while meteorological data are shown in Table 8. The main plots were assigned to a factorial combination of irrigation methods and regimes, while sub-plots were dedicated to the two sorghum cultivars, Speedfeed and Pegah. These cultivars have different genetic backgrounds and origins, with Speedfeed being a hybrid of sorghum–sudangrass and common sorghum from Australia [11] and Pegah (LFS56 × Early Orange) being an Iranian open-pollinated (OP) cultivar [12]. Two irrigation methods (drip and furrow, denoted as DRIP and FURW) were evaluated in this study, along with three irrigation regimes: full irrigation, moderate drought stress, and severe drought stress (supplying 100%, 75%, and 50% of the soil moisture deficit, denoted as I_100_, I_75_, and I_50_, respectively).

### 4.2. Land Preparation and Planting Operations

The seedbed was prepared by ploughing, harrowing, and leveling before planting. Based on the soil analysis and nutritional requirements of sorghum, the appropriate amount of fertilizer was incorporated into the soil. At the time of planting, 115 kg P_2_O_5_ ha^−1^ and 45 kg N ha^−1^ were applied from diammonium phosphate, while an additional 46 kg N ha^−1^ was supplied from urea fertilizer. Furthermore, 46 kg N ha^−1^ was added from urea fertilizer at the 4–6 leaf stage and also after the first cut harvest. The experiment consisted of 12 treatments and 36 experimental plots. Each sub-plot contained four planting rows that were 5 m in length with a spacing of 0.6 m between the rows. A plant density of 208,000 plants ha^−1^ was attained by planting at a distance of 8 cm between plants on the rows. The planting was conducted on 1 and 2 June in 2019 and 2020, respectively.

### 4.3. Irrigation

Soil moisture content was measured using a TDR device before each irrigation cycle to determine the required amount of irrigation water. The amount of water needed for the full irrigation treatment (I_100_) was estimated by subtracting the volumetric content of soil moisture from the field capacity point. Using Equation (1) [53], the volume of irrigation water required to restore soil moisture to the field capacity point was calculated based on the plot area and effective root depth:V_w_ = (θ_FC_ − θ_i_) × D × A(1)
where V_w_ is the volume of water used in each irrigation cycle for full irrigation treatment (m^3^), θ_FC_ is the volumetric content of soil moisture at the field capacity point (%), θ_i_ is the volumetric content of soil moisture before each irrigation (%), D is the effective vertical root depth (m), and A is the plot area (m^2^). The moderate (I_75_) and severe (I_50_) drought stress treatments used 75% and 50% of the full irrigation treatment water, respectively. Shut-off valves and volumetric meters were used to measure and control the amount of irrigation water used in each plot. In the furrow irrigation method, 5 cm polyethylene pipes directed water to the furrows, with four taps installed on the pipes within each plot to allow water to enter at 60 cm intervals. The drip irrigation method used strips with a 16 mm diameter and 10 cm dropper distance.

### 4.4. Harvesting and Measurements

At the conclusion of the vegetative growth stage, forages were harvested. To ascertain the green herbage yield, two middle rows of each plot were chosen for plant collection, with 0.5 m at both ends being discarded to eliminate any marginal effect. The leaves and stems were subsequently weighed separately. To determine the dry matter percentage, five plants were randomly selected, and their leaves and stems were dried separately in an oven set at 65 °C until weight stabilization. The dry matter yield was obtained by multiplying the dry matter percentage by the green herbage yield. The total green herbage and dry matter yields were finally calculated by summing the yields of two cuts. The IWUE was calculated using Equation (2) [2,21]:IWUE = DMY/WU(2)
where IWUE is irrigation water-use efficiency (kg m^−3^), DMY is the dry matter yield (kg ha^−1^), and WU is the total applied irrigation water (m^3^ ha^−1^). To determine the quality of forage, dried samples from each cut were ground and sifted through a 1 mm sieve. Subsequently, a mixed sample of milled forage from two cuts was prepared, taking into account the relative dry matter yield in each cut [4]. The crude protein content (CPC) was measured using the Kjeldahl method [54]. The neutral detergent fiber (NDF) and acid detergent fiber (ADF) were measured using a fiber analyzer [55]. Dry matter digestibility (DMD), dry matter intake (DMI), relative feed value (RFV), and net energy for lactation (NEL) were also calculated using Equations (3)–(6) [3,8,56].
DMD = 88.9 − (0.779 × ADF)(3)
DMI = 120/NDF(4)
RFV = DMD × DMI × 0.775(5)
NEL = [1.044 − (0.0119 × ADF)] × 2.205(6)

Metabolizable energy (ME) and organic matter digestibility (OMD) were measured by the in vitro gas production method [57] using Equations (7) and (8) [58,59]:ME = 2.20 + (0.136 × GP) + (0.057 × CP)(7)
OMD = 14.88 + (0.889 × GP) + (0.45 × CPC) + (0.0651 × CAC)(8)
where the ME is metabolizable energy (MJ kg^−1^), OMD is organic matter digestibility, GP is 24 h gas production (mL 200 mg^−1^), CPC is crude protein content, and CAC is crude ash content. Digestible dry matter yield (DDMY) and crude protein yield (CPY) were also calculated by multiplying dry matter yield (kg ha^−1^) by the DMD and CPC (g kg^−1^), respectively [8].

### 4.5. Statistical Analysis

The data underwent a combined analysis of variance, which was based on Bartlett’s test results and the homogeneity of experimental error variances observed over two years. The year was treated as a random effect, while the irrigation method, irrigation regime, and cultivar were treated as fixed effects. Statistical analysis was performed using SAS 9.1 software, and means were compared using the LSD method at a 5% probability level. To evaluate the relationship between traits and treatments, principal component analysis (PCA) was performed using XLSTAT software based on the correlation matrix of forage yield and quality, as well as the water-use efficiency of sorghum.

## 5. Conclusions

DRIP demonstrated higher forage yield and IWUE than FURW, with the superiority of DRIP increasing with an increase in drought stress intensity. Limited irrigation caused a decrease in forage yield and an increase in feed value for all irrigation methods and cultivars. The leaf-to-stem ratio and plant height were suitable indicators for comparing the quality and yield of sorghum forage, exhibiting a negative correlation between the quality and quantity of forage. Under I_100_ and I_75_, DRIP had higher forage quality, while FURW was superior in feed value under I_50_. Based on the findings, we recommend planting the cultivar Pegah and supplying 75% of the soil moisture deficiency through drip irrigation to achieve optimal yield and quality of sorghum forage while saving water. In severe water-limited conditions, it is advisable to plant the cultivar Pegah and supply 50% of the soil moisture deficiency through drip irrigation. If implementing drip irrigation is not possible, we recommend cultivating the hybrid Speedfeed and supplying 50% of the soil moisture deficiency through furrow irrigation.

## Figures and Tables

**Figure 1 plants-12-02154-f001:**
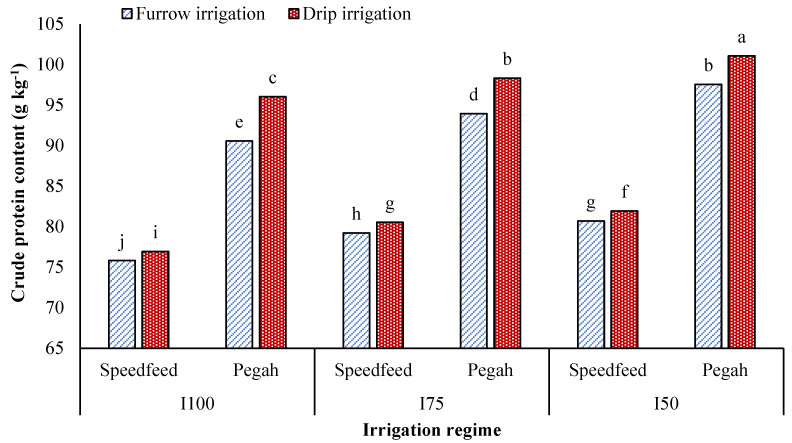
Effects of irrigation method × irrigation regime × cultivar on the crude protein content of sorghum. I_100_, I_75_, and I_50_, supplying 100%, 75%, and 50% of the soil moisture deficit, respectively. Different letters above the bars (means of replicates ± SE) indicate significant differences from each other at *p* < 0.05.

**Figure 2 plants-12-02154-f002:**
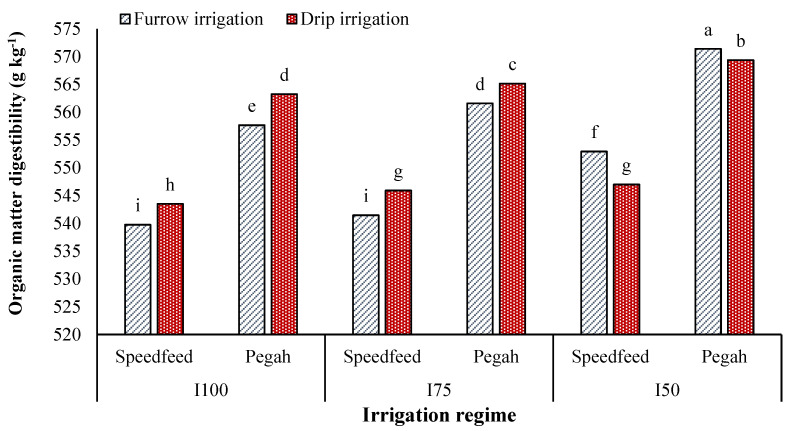
Effects of irrigation method × irrigation regime × cultivar on the organic matter digestibility of sorghum. I_100_, I_75_, and I_50_, supplying 100%, 75%, and 50% of the soil moisture deficit, respectively. Different letters above the bars (means of replicates ± SE) indicate significant differences from each other at *p* < 0.05.

**Figure 3 plants-12-02154-f003:**
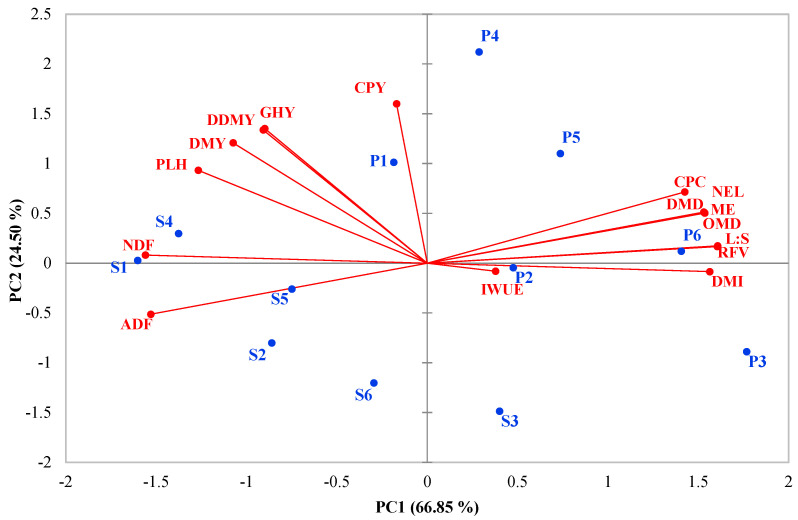
Principal component analysis (PCA) of the first two components performed on the forage yield and quality, and water-use efficiency of sorghum affected by irrigation method, irrigation regime, and cultivar; S: Speedfeed, P: Pegah, 1: Furrow + I_100_, 2: Furrow + I_75_, 3: Furrow + I_50_, 4: Drip + I_100_, 5: Drip + I_75_, 6: Drip + I_50_. GHY, green herbage yield; DMY, dry matter yield; IWUE, irrigation water-use efficiency; PLH, plant height; L:S, leaf-to-stem ratio; CPY, crude protein yield; DDMY, digestible dry matter yield; CPC, crude protein content; ADF, acid detergent fiber; NDF, neutral detergent fiber; DMD, dry matter digestibility; OMD, organic matter digestibility; DMI, dry matter intake; RFV, relative feed value; ME, metabolizable energy; NEL, net energy for lactation.

**Table 1 plants-12-02154-t001:** The main effects of year, irrigation method, irrigation regime, and cultivar on forage yield, water-use efficiency, and morphological characteristics of sorghum.

Experiment Factors	GHY ^†^	DMY	IWUE	PLH	L:S	CPY	DDMY
	(Mg ha^−1^)		(kg m^−3^)	(cm)		(kg ha^−1^)	
**Year**							
2019	99.60	21.94	5.246	193.0	0.607	1927	12,912
2020	103.62	23.01	5.374	195.0	0.609	1998	13,507
*p* Val.	0.223	0.178	0.386	0.478	0.791	0.159	0.220
LSD_0.05_	ns ^#^	ns	ns	ns	ns	ns	ns
**Irrigation method**							
Furrow	95.79	21.27	4.023	192.2	0.606	1818	12,463
Drip	107.43	23.68	6.597	195.8	0.610	2106	13,956
*p* Val.	0.047	0.046	0.004	0.277	0.219	0.017	0.038
LSD_0.05_	10.91	2.20	0.190	ns	ns	98	1123
**Irrigation regime**							
I_100_	121.97	26.50	4.498	216.2	0.587	2251	15,486
I_75_	101.57	22.58	5.136	197.9	0.607	1983	13,242
I_50_	81.31	18.35	6.296	167.8	0.630	1652	10,901
*p* Val.	0.001	0.001	0.001	0.001	0.010	0.001	0.001
LSD_0.05_	2.70	0.56	0.198	5.2	0.013	45	278
**Cultivar**							
Speedfeed	100.87	22.86	5.401	198.6	0.584	1805	13,178
Pegah	102.36	22.08	5.219	189.3	0.632	2119	13,241
*p* Val.	0.184	0.211	0.147	0.231	0.016	0.043	0.713
LSD_0.05_	ns	ns	ns	ns	0.015	268	ns

^†^ GHY, green herbage yield; DMY, dry matter yield; IWUE, irrigation water-use efficiency; PLH, plant height; L:S, leaf-to-stem ratio; CPY, crude protein yield; DDMY, digestible dry matter yield. ^#^ ns, non-significant.

**Table 2 plants-12-02154-t002:** The main effects of year, irrigation method, irrigation regime, and cultivar on forage quality of sorghum.

Experiment Factors	CPC ^†^	ADF	NDF	DMD	OMD	DMI	RFV	ME	NEL
	(g kg^−1^)						(%)	(Mcal kg^−1^)	
**Year**									
2019	88.1	384.8	581.9	589.2	555.4	20.65	94.34	2.008	1.292
2020	87.3	386.3	589.8	588.1	554.4	20.37	92.91	2.002	1.288
*p* Val.	0.554	0.609	0.024	0.610	0.569	0.027	0.047	0.584	0.608
LSD_0.05_	ns ^#^	ns	6.1	ns	ns	0.22	1.38	ns	ns
**Irrigation method**									
Furrow	86.3	386.7	585.6	587.7	554.1	20.52	93.53	2.000	1.287
Drip	89.1	384.4	586.1	589.6	555.7	20.50	93.73	2.010	1.293
*p* Val.	0.109	0.230	0.766	0.231	0.172	0.700	0.237	0.193	0.230
LSD_0.05_	ns	ns	ns	ns	ns	ns	ns	ns	ns
**Irrigation regime**									
I_100_	84.8	391.2	599.4	584.2	551.0	20.04	90.79	1.981	1.275
I_75_	88.0	387.9	586.6	586.8	553.5	20.47	93.15	1.996	1.284
I_50_	90.3	377.5	571.5	594.9	560.2	21.02	96.94	2.038	1.312
*p* Val.	0.014	0.002	0.001	0.002	0.001	0.001	0.001	0.001	0.002
LSD_0.05_	2.0	1.7	2.8	1.3	0.9	0.07	0.42	0.002	0.005
**Cultivar**									
Speedfeed	79.2	400.7	596.7	576.9	545.1	20.13	90.03	1.944	1.251
Pegah	96.2	370.5	575.0	600.4	564.7	20.89	97.22	2.066	1.330
*p* Val.	0.009	0.012	0.023	0.012	0.017	0.025	0.008	0.017	0.012
LSD_0.05_	3.1	7.4	10.1	5.8	6.6	0.38	1.07	0.042	0.019

^†^ CPC, crude protein content; ADF, acid detergent fiber; NDF, neutral detergent fiber; DMD, dry matter digestibility; OMD, organic matter digestibility; DMI, dry matter intake; RFV, relative feed value; ME, metabolizable energy; NEL, net energy for lactation. ^#^ ns, non-significant.

**Table 3 plants-12-02154-t003:** Effects of irrigation method × irrigation regime on forage yield, water-use efficiency, and morphological characteristics of sorghum.

Irrigation Method (IM)	Irrigation Regime (IR)	GHY ^†^	DMY	IWUE	PLH	L:S	CPY	DDMY
		(Mg ha^−1^)		(kg m^−3^)	(cm)		(kg ha^−1^)	
Furrow	I_100_	118.23	25.76	3.547	213.4	0.578	2140	14,968
	I_75_	95.41	21.31	3.912	196.8	0.604	1834	12,439
	I_50_	73.74	16.74	4.610	166.5	0.637	1480	9984
	LSD_0.05_	3.70	0.94	0.030	8.6	0.017	37	493
Drip	I_100_	125.70	27.24	5.449	219.0	0.596	2362	16,005
	I_75_	107.72	23.85	6.360	199.0	0.609	2132	14,045
	I_50_	88.87	19.95	7.981	169.2	0.624	1825	11,818
	LSD_0.05_	3.22	0.650	0.400	2.1	0.011	101	379
*p* Val.		0.033	0.046	0.008	0.045	0.014	0.045	0.049
LSD_0.05_ (IM × IR)		3.06	0.825	0.286	5.8	0.008	87	482

^†^ GHY, green herbage yield; DMY, dry matter yield; IWUE, irrigation water-use efficiency; PLH, plant height; L:S, leaf-to-stem ratio; CPY, crude protein yield; DDMY, digestible dry matter yield.

**Table 4 plants-12-02154-t004:** Effects of irrigation method × irrigation regime forage quality of sorghum.

Irrigation Method (IM)	Irrigation Regime (IR)	CPC ^†^	ADF	NDF	DMD	OMD	DMI	RFV	ME	NEL
		(g kg^−1^)						(%)	(Mcal kg^−1^)	
Furrow	I_100_	83.2	394.9	604.1	581.4	548.7	19.88	89.60	1.965	1.266
	I_75_	86.6	391.0	588.9	584.4	551.5	20.38	92.35	1.984	1.276
	I_50_	89.1	374.4	563.8	597.4	562.2	21.30	98.63	2.051	1.320
	LSD_0.05_	1.6	1.7	8.2	1.3	3.1	0.24	1.27	0.016	0.004
Drip	I_100_	86.5	387.6	594.8	587.1	553.4	20.20	91.98	1.996	1.285
	I_75_	89.4	384.9	584.2	589.2	555.5	20.56	93.95	2.009	1.292
	I_50_	91.5	380.6	579.3	592.5	558.2	20.73	95.25	2.025	1.303
	LSD_0.05_	2.3	2.1	3.8	1.5	1.4	0.17	0.62	0.012	0.006
*p* Val.		0.050	0.001	0.020	0.001	0.023	0.018	0.009	0.020	0.001
LSD_0.05_ (IM × IR)		0.06	1.1	8.2	0.8	3.2	0.27	1.28	0.019	0.003

^†^ CPC, crude protein content; ADF, acid detergent fiber; NDF, neutral detergent fiber; DMD, dry matter digestibility; OMD, organic matter digestibility; DMI, dry matter intake; RFV, relative feed value; ME, metabolizable energy; NEL, net energy for lactation.

**Table 5 plants-12-02154-t005:** Effects of irrigation method × cultivar on forage yield, water-use efficiency, plant height, and crude protein content of sorghum.

Irrigation Method (IM)	Cultivar (C)	GHY ^†^	IWUE	PLH	CPY	DDMY	CPC
		(Mg ha^−1^)	(kg m^−3^)	(cm)	(kg ha^−1^)		(g kg^−1^)
Furrow	Speedfeed	97.79	4.233	196.4	1742	12,800	78.6
	Pegah	93.80	3.814	188.0	1894	12,127	94.0
	LSD_0.05_	4.56	0.401	8.2	149	668	8.4
Drip	Speedfeed	103.94	6.569	200.9	1867	13,556	79.8
	Pegah	110.93	6.625	190.6	2345	14,356	98.5
	LSD_0.05_	4.59	ns ^#^	9.1	261	347	14.6
*p* Val.		0.049	0.046	0.048	0.029	0.049	0.048
LSD_0.05_ (IM × C)		4.47	0.310	8.4	131	559	12.3

^†^ GHY, green herbage yield; IWUE, irrigation water-use efficiency; PLH, plant height; CPY, crude protein yield; DDMY, digestible dry matter yield; CPC, crude protein content. ^#^ ns, non-significant.

**Table 6 plants-12-02154-t006:** Effects of irrigation regime × cultivar on forage yield, water-use efficiency, plant height, and feed value of sorghum.

Irrigation regime (IR)	Cultivar I	GHY ^†^	IWUE	PLH	CPY	DDMY	NDF	RFV
		(Mg ha^−1^)	(kg m^−3^)	(cm)	(kg ha^−1^)		(g kg^−1^)	(%)
I_100_	Speedfeed	118.45	4.454	221.5	2012	15,107	614.8	86.70
	Pegah	125.49	4.542	210.9	2489	15,865	584.1	94.88
	LSD_0.05_	6.98	ns ^#^	10.1	303	751	16.2	1.57
I_75_	Speedfeed	101.70	5.245	200.2	1851	13,330	594.8	90.00
	Pegah	101.43	5.027	195.7	2115	13,154	578.3	96.31
	LSD_0.05_	ns	0.211	ns	168	ns	10.1	3.30
I_50_	Speedfeed	82.44	6.503	174.2	1551	11,096	580.5	93.40
	Pegah	80.17	6.089	161.5	1754	10,705	562.6	100.48
	LSD_0.05_	2.21	0.013	11.5	92	172	16.7	1.70
*p* Val.		0.032	0.035	0.005	0.038	0.050	0.047	0.043
LSD_0.05_ (IR×C)		3.85	0.206	1.3	122	605	8.2	0.86

^†^ GHY, green herbage yield; IWUE, irrigation water-use efficiency; PLH, plant height; CPY, crude protein yield; DDMY, digestible dry matter yield; NDF, neutral detergent fiber; RFV, relative feed value. ^#^ ns, non-significant.

**Table 7 plants-12-02154-t007:** Physicochemical properties of soil at the experimental site.

Year	Texture	pH	OM (%)	N (%)	P (mg kg^−1^)	K (mg kg^−1^)	EC (dS m^−1^)	FC (%)	PWP (%)	AW (%)
2019	Clay loam	7.2	0.57	0.06	12.5	255	2.1	33	11	22
2020	Clay loam	7.1	0.58	0.07	12.7	249	2.0	32	10	22

OM, organic matter; EC, electrical conductivity; FC, field capacity (% volumetric moisture); PWP, permanent wilting point (% volumetric moisture); AW, available water (% volumetric moisture).

**Table 8 plants-12-02154-t008:** Temperature, evaporation, and precipitation during 2019 and 2020 growing seasons, at the experimental site.

Year	Month	T_mean_ (°C)	T_max_ (°C)	T_min_ (°C)	Evaporation (mm)	Precipitation (mm)
2019	June	27.58	35.02	19.10	297.1	11
	July	29.13	37.50	20.54	353.0	1
	August	26.72	34.84	18.83	299.6	8
	September	22.15	30.81	14.25	207.7	0
	October	16.84	23.52	11.22	130.8	76
2020	June	27.75	35.58	19.36	305.8	2
	July	29.57	38.28	20.15	372.2	20
	August	27.31	36.16	18.33	312.0	3
	September	22.82	31.51	14.30	221.7	0
	October	17.06	24.22	10.95	135.0	42

T_mean_, mean temperature; T_min_, minimum temperature; T_max_, maximum temperature.

## Data Availability

All data supporting the conclusions of this article are included in this article.

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
