# Peer review of "Irrigation Management Strategies to Enhance Forage Yield, Feed Value, and Water-Use Efficiency of Sorghum Cultivars"

_plants, 2023, doi:10.3390/plants12112154_

Round 1
Reviewer 1 Report
The study is well written and is clear in its objective. The experiments and research design are well organized and it delivers interesting information for the readers. I don’t have any major concerns for this manuscript and there are only a few suggestions in the introduction section that I highlighted in the annotated pdf.

Author Response
Dear Reviewer
We appreciate your comments on improving our article.
The study is well written and is clear in its objective. The experiments and research design are well organized and it delivers interesting information for the readers. I don’t have any major concerns for this manuscript and there are only a few suggestions in the introduction section that I highlighted in the annotated pdf.
Thanks for your comments. We revised our article based on your comments.
Best regards
Reviewer 2 Report
The paper is well-addressed, and the methodology is adequate for the objectives of the study. However, the text has many English errors and grammatical issues. Therefore, authors need to polish the whole manuscript with an English native speaker.
In addition:
This study evaluated Irrigation Management Techniques for Enhancing Forage Yield, Feed Value, and Water Use Efficiency of Sorghum Cultivars in Water-Stressed Environments. The methodology used is sufficient to the as of the study. The results are presented and discussed well. I have some suggestions below.
1-The title is too long. Please shorten it.
2-The abstract should be improved. Please add more results.
3-In the currentintroduction fits the content. However,authors should redraft this section explaining how this study should add significant improvement to the common knowledge.
4-The results section is written in a long way. Please revise this section and make it easily readable.
5-The conclusion section is too long. I recommend author condense this section and make it more precisely.
The text has many English errors and grammatical issues. Therefore, authors need to polish the whole manuscript with an English native speaker.
Author Response
Dear Reviewer
We appreciate your comments on improving our article.
The paper is well-addressed, and the methodology is adequate for the objectives of the study. However, the text has many English errors and grammatical issues. Therefore, authors need to polish the whole manuscript with an English native speaker.
Thanks a lot for your comments, We revised our article based on your comments and edited the English of our article by a native speaker.
In addition:
This study evaluated Irrigation Management Techniques for Enhancing Forage Yield, Feed Value, and Water Use Efficiency of Sorghum Cultivars in Water-Stressed Environments. The methodology used is sufficient to the as of the study. The results are presented and discussed well. I have some suggestions below.
1-The title is too long. Please shorten it.
Thanks a lot for your comments; we revised the article based on your comments.
2-The abstract should be improved. Please add more results.
Thanks a lot for your comments; we revised the article based on your comments.
3-In the currentintroduction fits the content. However,authors should redraft this section explaining how this study should add significant improvement to the common knowledge.
Thanks a lot for your comments; we revised the article based on your comments.
4-The results section is written in a long way. Please revise this section and make it easily readable.
Thanks a lot for your comments; we revised the article based on your comments.
5-The conclusion section is too long. I recommend author condense this section and make it more precisely.
Thanks a lot for your comments; we revised the article based on your comments.
Best regards
Reviewer 3 Report
Manuscript ID: plants-2378740
Title: Irrigation Management Techniques for Enhancing Forage Yield, Feed Value, and Water Use Efficiency of Sorghum Cultivars in Water-Stressed Environments
General remarks
The objective of the study was clearly stated, and the data are valuable for providing specific regional information on the influence of irrigation methods (drip and furrow irrigation), water deficit stress, and plant genetics and their interactions on yield, forage value, and irrigation water use efficiency of forage sorghum cultivars in a semi-arid region of Iran.
The topic is relevant to practical applications. It addresses a specific gap in the field.
The subject fits into the general area of Plants and is potentially of interest to the journal readership.
The title reflects the content. The abstract is sufficiently informative. The keywords are appropriate. The introduction is a brief overview of well-known concept. The objective is clear. The description of materials and methods is adequate. The statistical method is correct. The results are clearly presented. The conclusions are consistent with the evidence and arguments presented and address the main question posed. The organisation of the article is satisfactory. The tables and figures are necessary. The references are appropriate. The English language is understandable.
Therefore, I believe that the manuscript can be accepted for publication in Plants in present form.
Author Response
Dear Reviewer
We appreciate your comments on improving our article.
The objective of the study was clearly stated, and the data are valuable for providing specific regional information on the influence of irrigation methods (drip and furrow irrigation), water deficit stress, and plant genetics and their interactions on yield, forage value, and irrigation water use efficiency of forage sorghum cultivars in a semi-arid region of Iran.
The topic is relevant to practical applications. It addresses a specific gap in the field.
The subject fits into the general area of Plants and is potentially of interest to the journal readership.
The title reflects the content. The abstract is sufficiently informative. The keywords are appropriate. The introduction is a brief overview of well-known concept. The objective is clear. The description of materials and methods is adequate. The statistical method is correct. The results are clearly presented. The conclusions are consistent with the evidence and arguments presented and address the main question posed. The organisation of the article is satisfactory. The tables and figures are necessary. The references are appropriate. The English language is understandable.
Therefore, I believe that the manuscript can be accepted for publication in Plants in present form.
Thanks a lot for your comments; we revised the article based on your comments.
Best regards